



# Factors Causing Stratocumulus to Deviate from Subtropical High Variability on Seasonal to Interannual Timescales

Hairu Ding[1], Bjorn Stevens[1], and Hauke Schmidt[1]

[1]Max Planck Institute for Meteorology, Hamburg, Germany

**Correspondence:** Hairu Ding (hairu.ding@mpimet.mpg.de)

**Abstract.** Stratocumulus (Sc) covers the eastern flanks of maritime subtropical high pressure systems and exerts an influence on the global energy budget comparable to $CO_2$. Previous studies have identified the temperature difference between $700\,\mathrm{hPa}$ and the surface as the primary driver of Sc variability. However, the mechanistic linkages between subtropical highs and this critical temperature gradient, which defines lower tropospheric stability, remain unresolved. While subsidence modulates temperatures at $700\,\mathrm{hPa}$ and wind-driven cooling affects surface temperatures, the observed decoupling between subtropical high intensity and Sc fraction on seasonal to interannual timescales lacks a mechanical explanation. This study uses reanalysis data to test two hypothesized pathways linking the strength of the subtropical highs to the lower tropospheric stability. Results demonstrate that neither pathway dominates, as correlations between Sc-regime temperatures and subtropical high dynamics exhibit strong regional and temporal dependencies, indicating that correlation does not apply causation. Additionally, Sc-regime conditions do not systematically align with subtropical high variability, highlighting the need for further investigation into the dynamical processes governing temperatures in the lower troposphere.

## 1 Introduction

Stratocumulus (Sc), which covers around 20% of the low-latitude oceans, plays an important role in the global energy budget by reflecting solar radiation (Warren et al., 1986, 1988; Hahn and Warren, 2007; Wood, 2012). Previous studies have verified that even about 3.5–5% change in their fraction can lead to effects commensurate with those from a doubling of atmospheric $CO_2$ concentrations (Hartmann and Short, 1980; Randall and Suarez, 1984; Slingo, 1990), a fact that has motivated a considerable amount of research, including this study, to understand what controls Sc variations and changes.

To understand the variation of Sc, one approach has been to identify and study cloud controlling factors (Stevens and Brenguier, 2009). Previous researchers have made progressive efforts to define such factors statistically. For instance, already a century ago it was appreciated that Sc are sensitive to the temperature and humidity difference between the surface and air above the clouds (Blake, 1928). It is now understood that the sensitivity to the temperature difference arises because this difference measures the stability of the cloud-top interface, whereby greater stability suppresses the entrainment velocity, which limits the entrainment drying that inhibits cloud formation. Likewise for a given entrainment rate, a drier free atmosphere implies more entrainment drying. Klein and Hartmann (1993) made these relationships quantitative by demonstrating a strong and linear relationship between lower-tropospheric stability (LTS), which they defined as the difference between $700\,\mathrm{hPa}$ and




1000 hPa, and variations in Sc amount across regions and seasons. Later, Wood and Bretherton (2006) introduced the estimated inversion strength (EIS) to account for differences in lapse rates that influence the relationship between $\theta$ at 700 hPa and its more pertinent value, which lies just above cloud top. They found that EIS explains the variation of low clouds across a wider range of temperature regimes, and hence latitudes, to provide good predictions across tropical, subtropical, and mid-latitude regions. Kawai et al. (2017) combined EIS with the humidity difference into a unified index which they called estimated cloud top entrainment index (ECTEI). They tested ECTEI using ship observations and found it correlates with the total low stratiform clouds (including Sc, stratus, and sky-obscuring fog) better than EIS. Apart from the role of temperature and humidity differences, a physical understanding of Sc identifies a variety of other physical factors, including, large-scale atmospheric subsidence (Weaver and Pearson Jr, 1990; Randall and Suarez, 1984), downwelling longwave radiation above clouds (Stevens and Brenguier, 2009), and sea surface temperature (SST) (Klein et al., 1995).

Based on those factors, and from the spatial coincidence of Sc on the eastern-flanks of the semi-permanent subtropical high pressure regions, an expectation arises that variations in the strength of these high pressure regions will influence Sc. We identify two hypothesized pathways by which the strength of the subtropical highs may affect neighboring regions of Sc. The first hypothesized mechanism would be that variations in the strength of the high influence the free-tropospheric temperature above cloud top through enhanced adiabatic warming. This mechanism follows the pioneering work of Rodwell and Hoskins (2001), who demonstrate how monsoons can influence the strength of the high to their west, a phenomenon they termed the "monsoon-desert mechanism". The second hypothesized mechanism is that variations in the strength of the high change surface temperatures through its effect on the near surface wind and the wind's consequent effect on surface currents, upwelling, and surface cooling by wind-driven evaporation. Some evidence for such relationships exists based on studies that have found Sc to co-vary with the subtropical highs on synoptic (Klein et al., 1995; Klein, 1997; George and Wood, 2010; Toniazzo et al., 2011) and diurnal timescales (Ciesielski et al., 2001; Duynkerke and Teixeira, 2001; Garreaud and Muñoz, 2004). However, these relationships are not particularly strong. Moreover, there is scant evidence for such relationships on seasonal and interannual timescales, in which LTS dominants Sc variation (Klein and Hartmann, 1993; Wood and Bretherton, 2006; Richter and Mechoso, 2004, 2006). A few studies that have addressed this question have found the subtropical highs to be less important than the LTS itself (McCoy et al., 2017; Qu et al., 2015; Zhou et al., 2015). The extent to which this holds up to a more systematic analysis, and if so whether it is due to confounding influences, or simply by the fact that the strength of the high is a poor predictor of the constituent terms of cloud controlling factors as identified through a statistical analysis is the focus of the present study.

Specifically we test the two hypothesized pathways by which the strength of the subtropical high pressure areas may influence cloud amount in the main Sc areas. This requires defining the main Sc areas and the high pressure areas based on the data, as described in §2, and identifying whether the quantities hypothesized to be regulated by variations in the subtropical high are indeed the dominant cloud controlling factors. The different mechanisms by which variations in the high pressure regions can influence cloud controlling factors are tested in §3. The conclusions drawn from our analysis are presented in §4.





## 2 Data and Methods

### 2.1 Data

This paper uses ERA5 (the fifth generation ECMWF atmospheric reanalysis, E5, Hersbach et al. 2017) data for cloud controlling factors and atmospheric conditions. Its data is provided on a $0.25° \times 0.25°$ grid, with three-dimensional fields on 37 pressure levels. The monthly mean of mean sea level pressure, surface latent heat flux, wind components, vertical velocity, temperature, and geopotential heights are analyzed.

Low cloud fraction is analyzed based on the satellite data. This paper uses the second version of the ATSR-AATSR (Along-Track Scanning Radiometer and Advanced Along-Track Scanning Radiometer) data set in the Cloud_cci (European Space Agency Climate Change Initiative) project (Poulsen et al. (2017)). The data are provided on a $0.5° \times 0.5°$ grid. In this paper, Sc amount is denoted by the symbol $\kappa$ and defined to be equal to the low cloud fraction (cfc_low) in the identified Sc areas (see Section 2.2.2.1).

For the ATSR-AATSR data, monthly means for the period from January 2003 to December 2011 is analyzed and compared to ERA5 data over the same period. For the analyses of processes that impact the cloud controlling factors, the 30-year ERA5 record, from January 1985 to December 2014, is analyzed. To fix terminology, our use of the term "seasonal cycle" denotes the monthly climatology, while the term "interannual" denotes the variations in specific monthly values across the record—July for the Northern Hemisphere and January for the Southern Hemisphere. These months are chosen because subtropical highs are at their peak intensity, and Sc reaches its maximum during the summertime, when the monsoon-desert mechanism intensifies subsidence in their vicinity. We have also examined the interannual variability during the winter season, and the results are consistent with those observed during the summertime period. The "climatological mean" refers to the average over all months for the thirty-year record. Hence, for the thirty-year ERA data, at each spatial location, the seasonal cycle has 12 data points, the interannual record has 30-point records, and the climatological mean has one data point.

### 2.2 Definitions

A variety of quantities arise in our analysis and are defined as described as follows. In addition to defining the areas over which the analysis is performed, and the various cloud controlling factors being considered, two additional quantities are defined as possible pathways by which the strength of the subtropical highs might influence cloud-controlling factors and hence cloudiness, $\kappa$.

### 2.2.1 Marine subtropical highs and stratocumulus areas

Marine subtropical highs are defined by the union of closed $1020\,\mathrm{hPa}$ contours of the climatological-mean sea level pressure and marine areas. These are referred to subtropical high areas (H-areas) and denoted by color in Figure 1. Five major regions can be identified across the globe, and these are named, respectively, North Pacific (NP), North Atlantic (NA), South Pacific (SP), South Atlantic (SA), and South Indian Ocean (SI).





Similarly, Sc areas (c-areas) are defined when the mean low cloud fraction of 9 years is greater than 0.5 (or 0.4 for NA) and
falls within 45°N–45°S. c-areas are shown as thick black contours in Fig. 1. Figure 1 shows the c-areas are typically located
eastward of H-areas.

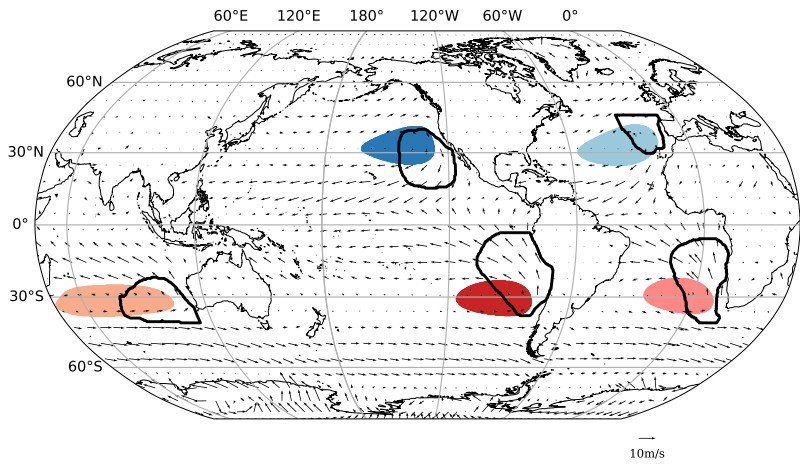

**Figure 1.** The defined H-areas (shaded) and c-areas (thick black lines). The quivers denote 10 m wind field. Each color represents one region,
and it is consistent in the later analyses. The map uses the Robinson projection.

This paper uses some similar words to address different concepts. "Regions" specifies the difference among NP, NA, SI,
SP, and SA. "Areas" specifies the difference between H- and c-areas. Subscripts "H" and "c" represent the area over which the
variables are averaged.

### 2.2.2    Cloud controlling factors

Previous studies suggested four factors to represent lower tropospheric conditions: the hydro-lapse ($\mathscr{H}$), LTS, EIS, ECTEI
(Klein and Hartmann, 1993; Wood and Bretherton, 2006; Kawai et al., 2017), as defined below:

$\mathscr{H}$ :  The hydro-lapse is defined as

$$\mathscr{H} = \beta \frac{l_\mathrm{v}}{c_p}(q_{700} - q_{1000}) \tag{1}$$

where $\beta = 0.23$, $l_\mathrm{v}$ is the latent heat of vaporization, $c_p$ is the specific heat of air at constant pressure, and $q$ is the specific
humidity. Here, and throughout, a numeric subscript denotes the pressure level in units of $\mathrm{hPa}$

LTS**:** The lower-tropospheric stability is defined as:

$$\mathrm{LTS} = \theta_{700} - \theta_{1000} \tag{2}$$

where $\theta$ is the potential temperature.



**EIS:** The estimated inversion strength is defined as:

$$\text{EIS} = \text{LTS} - \Gamma_{850} \left( Z_{700} - \text{LCL} \right) \tag{3}$$

where $\Gamma$ denotes the moist lapse rate, and $\Gamma_{850}$ is calculated by the average temperature of 700 and $1000\,\text{hPa}$. $Z$ denotes the geopotential height. LCL is the lifting condensation level, and it is fixed to be $500\,\text{m}$.

**ECTEI:** The estimated cloud top entrainment index is defined as:

$$\text{ECTEI} = \text{EIS} + \mathscr{H} \tag{4}$$

Some recent studies re-evaluated LTS and EIS and found that they have little difference in their ability to describe cloud amounts and that this difference varies with data sources (Cutler et al., 2022; Park and Shin, 2019). For this reason we re-examine the correlation between each suggested factor and $\kappa$ to select the one that is the best predictor. Figure 2 shows that

EIS best explains variations in $\kappa$ in our analyzed data, and it is selected in the later investigation. The performance of $\mathscr{H}$ is the worst. Even though there are some correlations between $\mathscr{H}$ and $\kappa$ in SP and SA, it still has the lowest correlation compared to the other factors in the same region. This agrees with Klein and Hartmann (1993), which claims that moisture dominates the change of stratus in the Arctic but not for subtropical Sc. For this reason (which through a supplementary analysis we also confirm, but don't include here) the effects of subtropical highs on Sc through the influence of their variations on the the free

tropospheric humidity are not explored further.

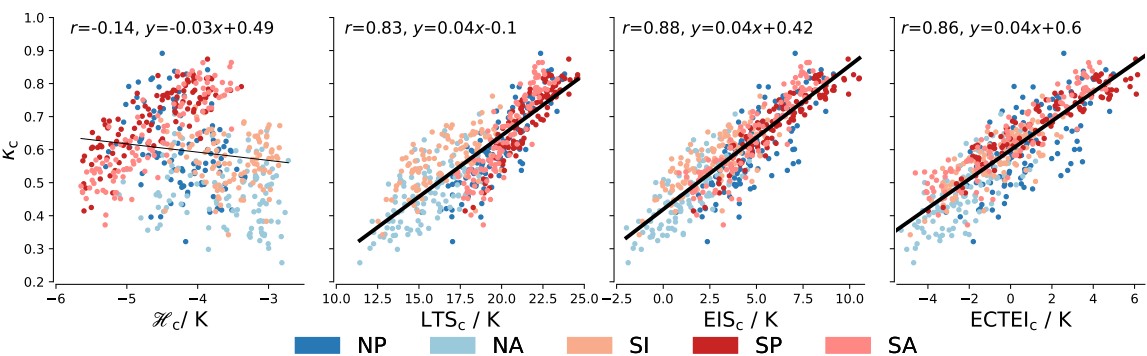

**Figure 2.** Scatter plots of different cloud controlling factors and low cloud fraction ($\kappa$) from 2003 to 2014. From left to right, the cloud controlling factors are $\mathscr{H}$, LTS, EIS, ECTEI, analyzed over the cloud (c) areas, as denoted by subscript. All subplots share the same y-axis, which represents $\kappa$, with each plot displaying data points colored by different regions. Regression lines are presented for p-values $\leq 0.05$, and thickened when $r^2 \geq 0.25$.

### 2.2.3 Adiabatic warming of lower-free troposphere

For a stronger high pressure we expect greater subsidence and more adiabatic warming, which in the mean would need to be balanced by increased radiative cooling or mean advection.





To explore the strength of the link between the lower-tropospheric potential temperature and the strength of the high pressure regions we look to the thermodynamic energy equation. There, assuming stationarity, adiabatic warming in $\mathrm{K/d}$ must be balanced by heating, $Q$, such that:

$$Q = \boldsymbol{v} \cdot \nabla\theta + \omega \frac{\partial\theta}{\partial p} \tag{5}$$

Here $\boldsymbol{v} \cdot \nabla\theta$ denotes horizontal advection, with $\boldsymbol{v}$, the horizontal wind vector, and $\omega\frac{\partial\theta}{\partial p}$ describes vertical advection, with $\omega$ representing vertical (pressure) velocity, and $p$ denoting the pressure. $Q$ can be associated with the convergence of radiant energy, or through turbulent mixing associated with covariances arising from the use of mean quantities to form the budget terms.

### 2.2.4 Wind driven surface cooling

A stronger high pressure associated with larger zonal geopolitical gradient ($\frac{\mathrm{d}\Phi}{\mathrm{d}x}$) is also expected to be accompanied by a cooler surface through a variety of mechanisms. First, it leads to more equatorward winds due to the geostrophic balance:

$$v = \frac{1}{f}\frac{\mathrm{d}\Phi}{\mathrm{d}x} \tag{6}$$

where $\Phi$ is the geopotential, $x$ represents distance in the zonal direction, $f$ is the Coriolis parameter, and we analyze $\frac{\mathrm{d}\Phi}{\mathrm{d}x}$ at $700\,\mathrm{hPa}$.

The consequent increased surface wind-stress leads to more ocean upwelling and hence surface cooling. This upwelling is measured by the Ekman pumping velocity, $w_\mathrm{E}$ :

$$w_\mathrm{E} = \nabla \times \left(\frac{\boldsymbol{\tau}}{\rho_0 f}\right). \tag{7}$$

Here $\rho_0 = 1030\,\mathrm{kg\,m^{-3}}$ is the density of ocean water, and

$$\boldsymbol{\tau} = \boldsymbol{v_{10}} \cdot \rho_\mathrm{a} C_\mathrm{D} \|V\| \tag{8}$$

is the surface wind-stress, with $\rho_\mathrm{a} = 1.225\,\mathrm{kg\,m^{-3}}$ the density of near-surface air, $C_\mathrm{D} = 0.0015$ the drag coefficient, $\boldsymbol{v_{10}}$ the near-surface ($10\,\mathrm{m}$) horizontal wind, and $\|V\|$ the near-surface wind speed. A positive value of $w_\mathrm{E}$ means upwelling motion, and a negative means downwelling motion. Different from other variables, the analyzed area of $w_\mathrm{E}$ is a continuous area of positive $w_\mathrm{E}$ near the coast.

In addition, increased surface winds can also cool the surface through enhanced evaporation, which is measured by the latent heat flux (LHF).

## 3 Analysis

In this section we first explore what factors explain variations in EIS, which we now take as a proxy for cloudiness $\kappa$. We then explain to what extent these factors can be related to variations in the strength of the subtropical high pressure regions.





### 3.1 Dependence of EIS on $\theta_{700}$ and $\theta_{1000}$

EIS differs from LTS as it includes the temperature-dependent lapse rate $\Gamma$, but it is dominated by the variations of LTS (i.e., the difference between $\theta_{700}$ and $\theta_{1000}$) because any change in lapse rates depends on the change of the temperature below

700 hPa. Table 1 shows how these quantities vary across different regions and for different timescales and how much they contribute to variability in EIS.

In the higher latitude regions of the NP, NA and SI, variations of $\theta_{700}$ are mostly larger than variations in $\theta_{1000}$ in both the seasonal and interannual data and explain most of the variability in EIS in those regions and on those timescales. In these regions $\theta_{700}$ and $\theta_{1000}$ strongly co-vary across the seasonal cycle, but $\theta_{700}$ varies more. This means that EIS increases even as

$\theta_{1000}$ increases, which explains the otherwise counterintuitive positive correlation between $\theta_{1000}$ and EIS in the higher-latitude regions on seasonal timescales. For the more equatorward regions of the SP and SA, variability of EIS is dominated by $\theta_{1000}$, of which the variability much larger than that of $\theta_{700}$ on seasonal timescales. Variations in $\theta_{1000}$ are important for all regions on interannual timescales (as evidenced by the correlations between $\theta_{1000}$ and EIS in Table 1), and are particularly important for the main Sc areas of the NP, SP and SA, where the standard deviations of $\theta_{1000}$ and $\theta_{700}$ are nearly equal.

| c-area | Seasonal Cycle | | | | | | Interannual Variability | | | | |
|--------|----------------|---|----------------|---|---------------------|---|-------------------------|---|----------------|---|---------------------|
| | $\theta_{1000,c}$ | | $\theta_{700,c}$ | | $\omega_{700,H}$ | | $\theta_{1000,c}$ | | $\theta_{700,c}$ | | $\omega_{700,H}$ |
| | $r$ | $\sigma/K$ | $r$ | $\sigma/K$ | $\sigma/hPa\,d^{-1}$ | | $r$ | $\sigma/K$ | $r$ | $\sigma/K$ | $\sigma/hPa\,d^{-1}$ |
| NP | 0.16 | 1.5 | **0.64** | **2.6** | 10.3 | | **-0.83** | 0.6 | 0.76 | **0.7** | 5.4 |
| NA | 0.26 | 2.4 | **0.49** | **3.5** | 5.3 | | -0.53 | 0.5 | **0.75** | **0.8** | 3.2 |
| SI | 0.52 | 1.8 | **0.79** | **3.3** | 8.1 | | **-0.84** | 0.7 | 0.72 | 0.7 | 5.4 |
| SP | **-0.97** | **2.0** | -0.73 | 1.0 | 4.4 | | -0.48 | 0.4 | **0.85** | **0.9** | 6.9 |
| SA | **-0.94** | **2.0** | -0.52 | 1.0 | 3.7 | | **-0.66** | 0.4 | 0.64 | **0.5** | 4.9 |

**Table 1.** Correlation ($r$) between $\theta_{1000,c}$, $\theta_{700,c}$ and EIS; and the standard deviation ($\sigma$) for $\theta_{1000,c}$, $\theta_{700,c}$, and subsidence rate ($\omega_{700,H}$) over the seasonal cycle and the interannual variability. The dominant contribution to EIS is denoted by a bold font.

This analysis demonstrates that variations in EIS are complex and regionally dependent. It also hints at why previous studies have not reported a strong relationship between the strength of the subtropical highs and cloudiness, as might be expected if, for instance, cloudiness was primarily controlled by either variations in the temperatures at the surface, driven by surface winds and upwelling, or aloft, driven by downwelling. Using $\omega_{700,H}$ as a measure of the strength of highs does not clearly indicate whether the variability of $\theta_{700}$ is larger or smaller than that of $\theta_{1000}$. For the seasonal cycle, the three regions with the largest

variability of the strength of the high (measured by the standard variation of $\omega_{700,H}$ in Table 1) are also the regions where $\theta_{700}$ varies more than $\theta_{1000}$. However, this correlation does not imply causation, as the interannual series fails to exhibit the same relationship.

To explore whether such relationships might exist, but are hidden by co-variability in other factors, we regress EIS against $Q_c$, the adiabatic warming in the cloud region, as well as two fields that could be indicative of the influence of the subtropical





high on surface processes, one being the $\text{LHF}_\text{c}$, which we expect to co-vary with the surface wind speed, $\|V\|$, the other being

ocean upwelling, $w_\text{E}$. Fig.3 shows that some weak relationships emerge, as expected. Greater adiabatic warming, stronger

winds (as measured by surface-latent heat fluxes) and more upwelling all are positively correlated with increases in EIS. The

somewhat weaker relationship between EIS and LHF can be expected because while surface winds cool the surface, and hence

lower $\theta_{1000}$, increased latent heat fluxes also breakup the cloud decks (Bretherton and Wyant, 1997). Because the relationships

are generally stronger over the seasonal cycle as compared to the interannual record, it raises the question as to whether they

are causal. We investigate this question in more depth below by separately investigating how $\theta_{700}$ varies with the heat budget,

and how LHF and $w_\text{E}$ varies through the surface momentum budget, as might be expected if they were respectively controlled

by variations in the strength of the neighboring high pressure regions.

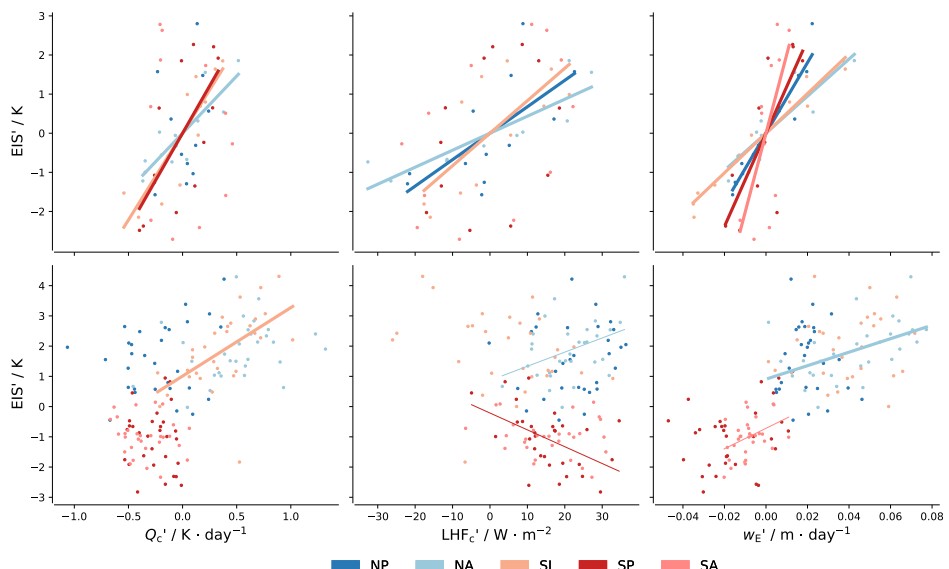

**Figure 3.** Scatter plots of $Q$ and EIS (left), LHF and EIS (middle), and $w_\text{E}$ and EIS (right). Each color represents a region. The top branch is for the seasonal cycle, and the bottom branch is for the interannual time series. Regression lines are presented for p-values $\leq 0.05$, and thickened when $r^2 \geq 0.25$.

## 3.2 Control by atmospheric downwelling

Before exploring to what extent variations in the strength of the high pressure regions can explain variations in $\theta_{700}$ we first

examine the more basic question as to whether the strength of the adiabatic warming, which in stationarity balances the diabatic

cooling $Q$, co-varies with $\theta_{700}$ in the Sc areas. Fig. 4 demonstrates that there is a clear relationship between $Q_\text{c}$ and $\omega_{700,\text{c}}$ in

the climatological mean (Fig. 4). A strong and consistent relationship also emerges across the seasonal cycle and at interannual

timescales (Fig. 5). Hence $\omega_{700,\text{H}}$ is a good proxy for the strength of the adiabatic warming in the cloud areas.





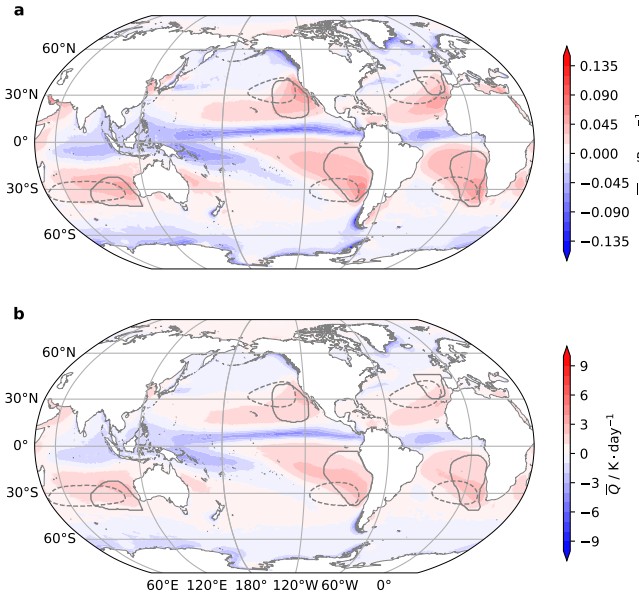

**Figure 4.** Map of 1985–2014 climatological mean $\omega_{700}$ (left) and $Q$ (right). Stratocumulus regions are shown by dashed lines and cloud-regions are outlined by solid lines.

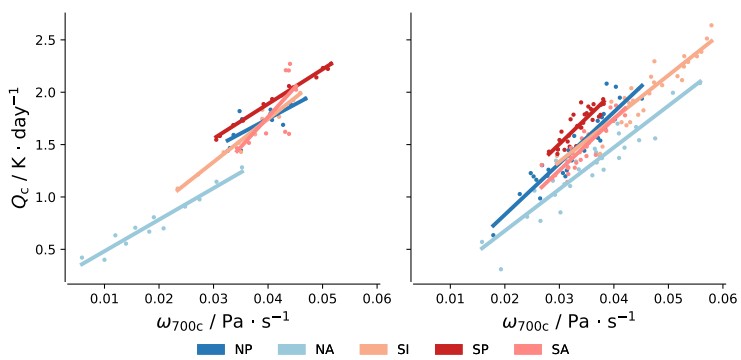

**Figure 5.** Scatter plots of $\omega_{700}$ and $Q$. Each color represents a region. The left subplot is for the seasonal cycle, and the right subplot is for the interannual time series. Regression lines are presented for p-values $\leq 0.05$, and thickened when $r^2 \geq 0.25$.





Knowledge of the adiabatic warming is, however, not sufficient to determine $\theta_{700}$, as might naively be expected from a monsoon-desert mechanism following the arguments of Rodwell and Hoskins (2001). This is shown in Fig. 6, which shows no consistent relationship between $Q_{\mathrm{c}}$ and $\theta_{700,\mathrm{c}}$ across the seasonal cycle for the different regions, and no relationship between $Q_{\mathrm{c}}$ and $\theta_{700,\mathrm{c}}$ whatsoever on interannual timescales. This indicates that $\omega_{700,\mathrm{c}}$ is not the dominant factor for changes in $\theta_{700}$, and falsifies the hypothesis that variations in $\theta_{700,\mathrm{c}}$ can be explained by a monsoon-desert mechanism, at least one mediated

by the strength of the adiabatic warming.

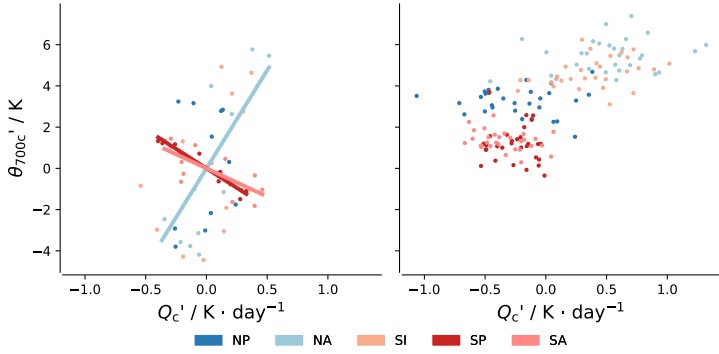

**Figure 6.** Scatter plots of $Q$ and $\theta_{700}$. Each color represents a region. The left subplot is for the seasonal cycle, and the right subplot is for the interannual time series. Regression lines are presented for p-values $\leq 0.05$, and thickened when $r^2 \geq 0.25$.

### 3.3   Control by wind-driven cooling

The strengthening or weakening of subtropical highs is associated with not only the changed subsidence rate but also the changed geopotential gradient and near-surface wind. The geopotential gradient depends on both the structure and the strength of the subtropical highs. According to Sverdrup balance, the equatorward near-surface wind, which in geostrophic balance

is determined by the zonal geopotential gradient ($\frac{\mathrm{d}\Phi}{\mathrm{d}x}$), is associated with a wind-stress gradient that results in ocean Ekman pumping (Anderson and Gill, 1975). In addition, the changed near-surface wind can also affect cold advection of waters from high-latitudes, and impact surface evaporative cooling as measured by the surface latent heat flux (LHF).

     Figure 7 shows patterns for mean $\frac{\mathrm{d}\Phi}{\mathrm{d}x}$, Ekman pumping velocity ($w_{\mathrm{E}}$), and LHF. Unlike $Q$ and $\omega_{700}$, pattern correlations are difficult to discern. Upwelling areas are restricted to the coastal regions where the wind-stress curl is large, and the maximum

LHF is located on the west and equatorward side of the maximum $\frac{\mathrm{d}\Phi}{\mathrm{d}x}$ where temperatures are warmer and strong trade-winds prevail. A more quantitative evaluation of the relationship between $\frac{\mathrm{d}\Phi}{\mathrm{d}x}$ and either $w_{\mathrm{E}}$ or LHF (not shown), does not show strong and consistent relationships across regions or timescales. This leads us to conclude that variations in near-surface geopotential gradients are not the primary driver of changes in $\theta_{1000}$ and hence variations in EIS.



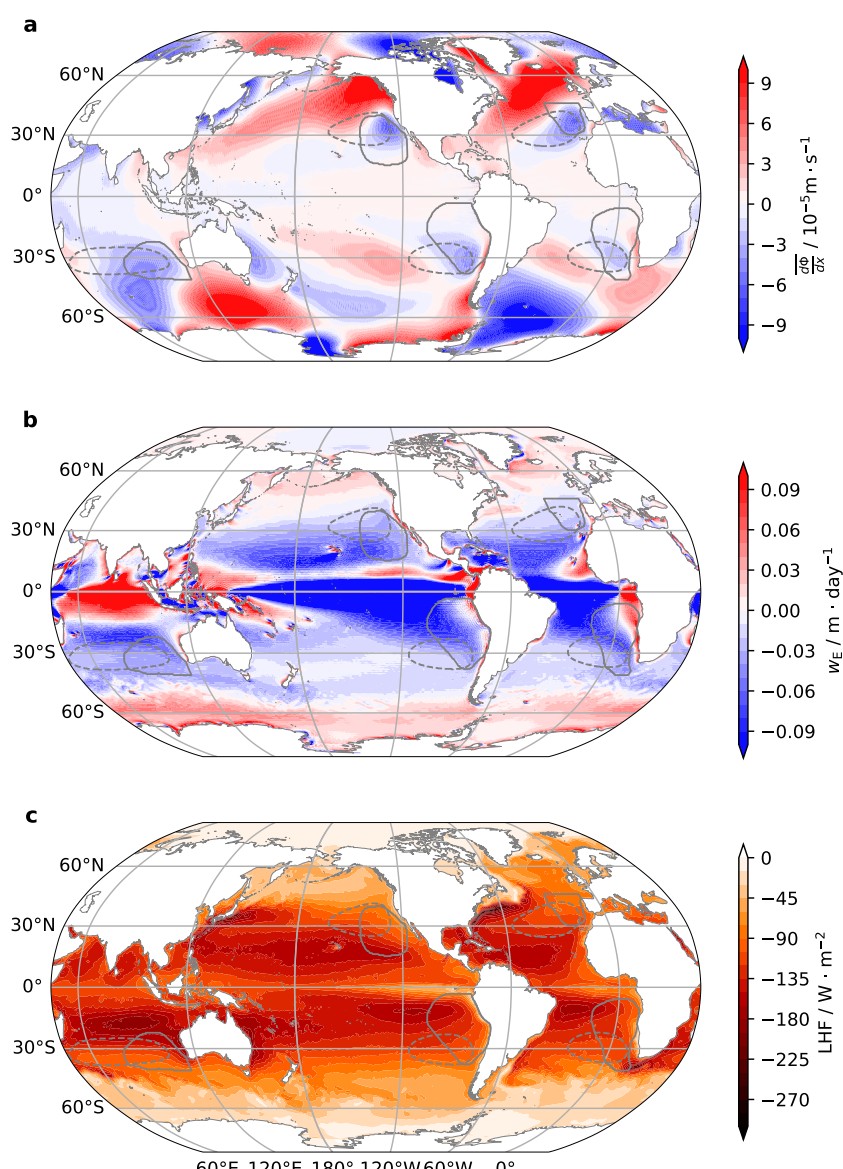

**Figure 7.** Map of 1985–2014 climatological mean a. $\frac{\mathrm{d}\Phi}{\mathrm{d}x}$, b. $w_{\mathrm{E}}$, c. LHF.





### 3.4 The gap between changes in H- and c- areas

Until this point we have considered proxies within the cloud areas for the strength of the high pressure areas that generally lie westward of the cloud areas. Figure 8 shows that cloud-area measures of the high pressure regions are not necessarily good proxies for the high pressure areas, as no rule for the relationships between H- and c- areas emerges. Correlations, when they exist can differ in sign across regions and for the same region across timescales. Even though the regression coefficients between the two areas show some similarity in NA across timescales, the correlations in the interannual time series are weak.

Therefore, the properties of c-areas do not simply follow the properties of H-areas, which further reduces the probability of predicting Sc by the strength and structure of subtropical highs.

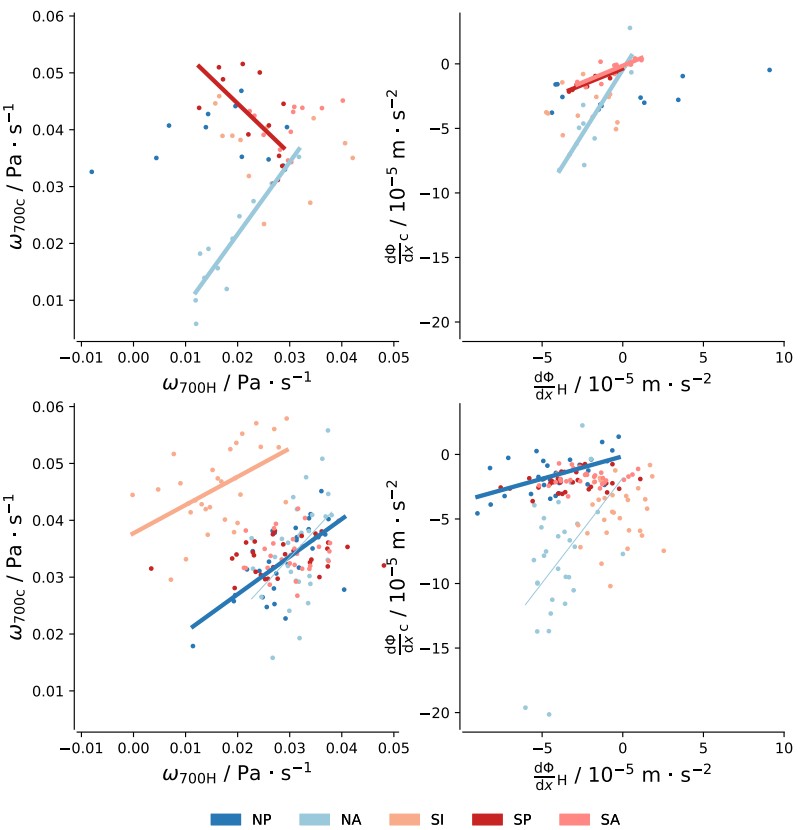

**Figure 8.** Scatter plots of $\omega_{700}$ (top) and $\frac{\mathrm{d}\Phi}{\mathrm{d}x}$ (bottom) between H- and c- areas. Each color represents a region. The top branch is for the seasonal cycle, and the bottom branch is for the interannual time series. Regression lines are presented for p-values $\leq 0.05$, and thickened when $r^2 \geq 0.25$.





## 4 Conclusions

This paper tests the two hypothesized processes by which subtropical highs may affect Sc. Both hypotheses are rejected.

First we defined the marine high pressure areas and the Sc areas by a surface isobar in the case of the former and a satellite
derived cloud faction in the latter. This identified five regions, in each of which the high pressure area intersects the cloud areas, with the latter generally found on the eastward flanks of the former. Next we demonstrated that Sc is well predicted by both the lower-tropospheric stability (LTS) and the Estimated Inversion Strength (EIS). A more recent proposal for a cloud controlling factor, the Estimated Cloud Top Entrainment Index, is more complex as it includes humidity variations, and performed slightly worse than the EIS, as its additional skill arises from predictions outside of the main cloud areas we consider. Furthermore, in
the areas we considered, and for any particular region, most of the variability in EIS can be explained by the lower-tropospheric stability alone.

Given these findings we hypothesized that a monsoon-desert mechanisms, which is known to strengthen regions of climatological high pressure, could lead to increased cloudiness if the EIS (or LTS) increased with the strength of the high pressure regions, through an increase in adiabatic warming, $Q$, that maintains a higher potential temperature at $700\,\mathrm{hPa}$. However, we
found that the variation of $Q$ is not the dominant factor for the cloud-top temperature ($\theta_{700}$) change, as the response of $\theta_{700}$ can behave oppositely in different regions. This agrees with (Caldwell and Bretherton, 2009), which shows that the effects of the thermodynamic process are not warming the cloud-top directly; instead, it works more on enhancing the subsidence itself, which all things considered would maintain a shallower cloud layer.

We further hypothesized that the strength of the subtropical highs could modulate cloud amounts through their effect on the
235 surface momentum balance, and hence surface temperature, to which $\theta$ at $1000\,\mathrm{hPa}$ is strongly related. A strengthened high is posited to increase surface wind speeds, which lead to more cold-water advection, greater surface evaporation and cooling and more upwelling. If this mechanism was a dominant factor in controlling cloud amount, we hypothesized that there should be a strong relationship between the strength of the surface geopotential gradient and the latent-heat flux, and the surface upwelling as measured by the wind-stress curl. Our analysis found little evidence of such a relationship.

Environmental changes associated with variations in the strength of the subtropical high pressure regions correlate better with LTS (and EIS) than they do with the components of the LTS and EIS the variations are thought to influence. At both seasonal and interannual timescales regionally unified positive correlations between $Q$ and EIS, as well as between upwelling velocity and EIS can be identified. We interpret this as indicative of variations in high pressure regions not being the primary cause of variations in the $Q$, $w_{\mathrm{E}}$ or LHF, but rather indicative of hidden processes that cause these quantities to co-vary with
LTS. This finding is further supported by the lack of robust relationships between the variations in $Q$, $w_{\mathrm{E}}$ or LHF, in the high pressure region and the same quantity in the partially overlapping cloud regions.

Our results unfortunately do not support the hypothesis that an understanding of how the subtropical highs change with climate will be informative for how Sc amount will change in regions where such clouds prevail. It is well appreciated that the temperature control of the convecting areas on the moist adiabatic lapse rate throughout the tropics can influence the
250 near tropical LTS (Manabe et al., 1965; Stone and Carlson, 1979; Betts, 1986; Mapes, 1993; Wood and Bretherton, 2004;



Schneider, 2007). However, there is also a growing appreciation of the departures from the weak-temperature gradients that this mechanism relies on, which will continue to motivate efforts to identify dynamic factors influencing the LTS across the near tropics (Sobel and Bretherton, 2000; Sobel et al., 2001; Singh and O'Gorman, 2013; Bao and Stevens, 2021).

*Data availability.* The datasets used in this work can be obtained from the ERA5 and ATSR-AATSR online repositories at https://www.
ecmwf.int/en/forecasts/datasets/reanalysis-datasets/era5 and http://catalogue.ceda.ac.uk/uuid/1ea3b2e391e4441daa57100a02b98691.

*Author contributions.* HD did the data process and analyses, and wrote the original draft of this paper and edited it later. BS initialized the idea and hypotheses, restructured this paper and wrote analyses. HS reviewed and edited this paper.

*Competing interests.* The authors declare that no competing interests are present.

*Acknowledgements.* The German Climate Computation Centre (DKRZ) provided the reanalysis data and supercomputer used in this study.
The authors appreciate their support for the data pool and server. The authors also appreciate Hans Segura, Tiffany Shaw, Nedjeljka Zagar, Ian Dragaud, and Marco Giorgetta, who shared their thoughts that influenced this paper.



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
