# Peer review of "Factors Causing Stratocumulus to Deviate from Subtropical High Variability on Seasonal to Interannual Timescales"

_EGUsphere, 2025_

## Referee Comment (RC1)

The topic of this manuscript is very interesting and I am personally very curious about this topic. The results look reasonable. I enjoyed reading the manuscript.

The idea of the analysis is nice and the figures are very clear and easy to understand.

I have some comments and questions.

**Major comments:**

0. First, a common index for subtropical high is omega at 700hPa? I imagined the most common index is the sea surface pressure (SLP).

1. The title says "Factors Causing Stratocumulus to Deviate from Subtropical High Variability". So I imagined the ultimate target is the relationship between the low cloud fraction and SLP. If so, I wanted to see the relationship between them for the seasonal and interannual time scales first. In addition, I guess this relationship (low cloud fraction and SLP) can be decomposed into the following relationships.

    (A) the low cloud fraction and EIS.              x   (for seasonal and interannual)

    (B) EIS and $\theta 700$                              o
    (C) $\theta 700$ and Q (defined by eq. (5))              o
    (D) Q and $\omega 700$                              o
    (E) $\omega 700$ and SLP                              x

    (F) EIS and $\theta 1000$                              o
    (G) $\theta 1000$ and ocean upwelling              x
    (H) $\theta 1000$ and latent heat flux              x
    (I) ocean upwelling and gradient of SLP              o
    (J) latent heat flux and gradient of SLP              o
    (K) a gradient of SLP and SLP itself              x

The authors showed the important essential relationships among them (Relationships marked by "o" are (at least partially) shown. Relationships marked by "x" are not

shown.). However, could the authors schematically describe in the text which relation is discussed in the present paper (because it is a bit hard to catch what is discussed and what is not discussed in the manuscript.)? The relationships that are not shown could also be helpful for the readers. Could the authors add such figures (at least (A) for seasonal and interannual, and (E).) in supplemental materials? I also want to know the correlation between EIS and omega700 (that is not shown in Table 1).

1'. (Related to item 1.) Could the authors clarify the reason why θ1000 are not examined at the beginning of the analysis in the text more explicitly if the authors don't do them? Is this because "Environmental changes associated with variations in the strength of the subtropical high pressure regions correlate better with LTS (and EIS) than they do with the components of the LTS and EIS the variations are thought to influence (L240-241)"?

**Minor comments:**

L1-2:   "Stratocumulus (Sc) covers the eastern flanks of maritime subtropical high pressure systems and exerts an influence on the global energy budget comparable to CO2."
This sentence seems to be unclear. Do the authors mean the total CO2 radiative effect (difference between the current CO2 concentration level and zero CO2)? Or the change of (doubling) CO2 (~4 W/m2)? The influence of Sc is (the global average of) cloud radiative effect of Sc itself (difference between the current Sc existence and clear state. not change in CRE of Sc)?

L4:   "gradient"
"difference" is better?

L39-41: "The first hypothesized mechanism would be that variations in the strength of the high influence the free-tropospheric temperature above cloud top through enhanced adiabatic warming. This mechanism follows the pioneering work of Rodwell and Hoskins (2001), who demonstrate how monsoons can influence the strength of the high to their west"
The first sentence mentions the influence of high-pressure systems on clouds (the cause is the variation in high-pressure systems and the effect is the variation in clouds). The second sentence seems to discuss the influence of the

monsoons on the variation of high-pressure systems (the cause is the monsoon variations and the effect is the variations of high-pressure systems.). Are the mechanisms similar?

L157-164 (2nd paragraph of 3.1):

I thought the essence of these characteristics was discussed in Klein & Hartmann (1993). So it's appropriate to cite the publication here.

L168:    "or aloft, driven by downwelling"

What is "downwelling"? Air (atmospheric) subsidence? Or ocean downwelling? Could you please clarify what downwelling or upwelling you mean when you use these terms?

L169-171:

"For the seasonal cycle, the three regions with the largest variability of the strength of the high (measured by the standard variation of $\omega_{700,H}$ in Table 1) are also the regions where $\theta_{700}$ varies more than $\theta_{1000}$. However, this correlation does not imply causation, as the interannual series fails to exhibit the same relationship."

Especially mid-latitude (including subtropics), the seasonal cycle of $\theta_{700}$ is mainly controlled by the seasonal cycle of solar zenith angle (not by omega700), I thought. I guess people generally imagine that $\omega_{700}$ cannot determine $\theta_{700}$ (at least) except tropics. Am I wrong?

L177-179:

"The somewhat weaker relationship between EIS and LHF can be expected because while surface winds cool the surface, and hence lower $\theta_{1000}$, increased latent heat fluxes also breakup the cloud decks"

This sentence discusses the relationship between EIS and LHF. So the fact that increased latent heat fluxes break up the cloud decks cannot be the reason for the relationship between EIS and LHF (i.e., cloud cover doesn't affect EIS directly), I guess.

L184:    "atmospheric downwelling"

Does this mean the air (atmospheric) subsidence?

Captions of Figure 3 and 6:

Plotted variables look not raw values but anomalies of the variables. In addition, I couldn't find the meaning of the apostrophe attached to the variables.

L230:    "cloud-top temperature"

L232:    cloud-top

Generally, 700 hPa is not cloud top but above stratocumulus and in the free atmosphere, I guess.

L242:    "upwelling velocity"

Can you clarify what upwelling this is?   Ocean upwelling velocity?

---

## Author Comment (AC1)

Please note that the line numbers referenced in this response refer to **the version of the manuscript with tracked changes**, as they may differ from the ones in the clean version.

**Response to Referee #1**

**Major comments**

0. First, a common index for subtropical high is omega at 700hPa? I imagined the most common index is the sea surface pressure (SLP).

Yes, SLP is indeed a common index for subtropical highs, and we use it to identify the climatological location of subtropical highs (Section 2.2.1). However, in this study, we use $\omega_{700}$ and $d\Phi/dx$ in the Sc area instead of SLP in the subtropical high area, based on our hypothesized free-troposphere and surface pathways. Our updated Section 3.4 adds the relationships between SLP and the two selected factors to show the deviation between the two factors and SLP. Apart from that, we still interested in whether the hypothesized pathways work in the Sc area. Hence, we use $\omega_{700}$ and $d\Phi/dx$ in the Sc area to ensure the physical interpretability of the results.

By this comment we realized that the phrase "strength of subtropical highs" might lead readers to associate it with SLP, rather than the subsidence rate or geopotential gradient that we intended to convey through the two pathways. Hence, we've revised the manuscript to avoid using "strength/intensity" in this context. An additional explanation is added in L59-65.

1. The title says "Factors Causing Stratocumulus to Deviate from Subtropical High Variability". So I imagined the ultimate target is the relationship between the low cloud fraction and SLP. If so, I wanted to see the relationship between them for the seasonal and interannual time scales first. In addition, I guess this relationship (low cloud fraction and SLP) can be decomposed into the following relationships.

| | | |
|---|---|---|
| (A) | the low cloud fraction and EIS | x (for seasonal and interannual) |
| (B) | EIS and θ700 | o |
| (C) | θ700 and Q (defined by eq. (5)) | o |
| (D) | Q and ω700 | o |
| (E) | ω700 and SLP | x |
| (F) | EIS and θ1000 | o |
| (G) | θ1000 and ocean upwelling | x |
| (H) | θ1000 and latent heat flux | x |
| (I) | ocean upwelling and gradient of SLP | o |
| (J) | latent heat flux and gradient of SLP | o |
| (K) | a gradient of SLP and SLP itself | x |

The authors showed the important essential relationships among them (Relationships marked by "o" are (at least partially) shown. Relationships marked by "x" are not shown.). However, could the authors schematically describe in the text which relation is discussed in the present paper (because it is a bit hard to catch what is discussed and what is not discussed in the manuscript.)? The relationships that are not shown could also be helpful for the readers. Could the authors add such figures (at least (A) for seasonal and interannual, and (E).) in supplemental materials? I also want to know the correlation between EIS and omega700 (that is not shown in Table 1).

Thanks a lot. We agree that the listed decomposition of relationships is useful. A description of which relation is discussed is added at the end of the introduction. (L69-75)

All the relationships listed above have been analyzed:

(A) is shown in the manuscript only for all monthly means (Fig. 2), but we've added the information for the seasonal and the interannual timescales. Corresponding plots are shown below. We've also added a sentence noting the inclusion of these correlation coefficients in L141-142.

(B)-(D) are already shown in Table 1 and Figures 4-6.

(E) has been added as a replacement for the right column of Figure 9. The revised Figure 9 shows that $\omega_{700}$ in the Sc area deviates from that in the subtropical high area, and $\omega_{700}$ in the high area covaries with SLP there at different rates. Hence, the simple factor of SLP in subtropical highs cannot represent subsidence in the Sc area.

(F) is analyzed in Table 1.

(G)-(H) are added as the new Figure 8.

(I)-(J) are shown below but are not put in the manuscript as they provide the same information as the current Figure 8—-that the hypothesized surface pathway is rejected.

(K) is added as Figure 10, describing that the geopotential gradient in the Sc area doesn't covary with SLP in the subtropical high area.

[Figure]

Plots for (A): scatter plots of low cloud fraction ($\kappa$) vs. EIS on the seasonal timescale (left) and on the interannual time scale (right). Correlation coefficients are represented by $r$ and the regression lines (black) are presented for $p-values \leq 0.05$. Equations of regression lines are shown besides correlation coefficients. Each colour represents one region.

[Figure]

Plots for (I)-(J): scatter plots of geopotential gradient and latent heat flux (left) and of geopotential gradient and ocean upwelling velocity (right). The primes indicate deviations from the mean of the respective regions on the corresponding timescales. The top branch is for seasonal and the low branch is for interannual series. Regression lines are presented for $p-values \leq 0.05$, and thickened when $r^2 \geq 0.25$.

The correlation between EIS and ω700 has been added in Table 1.

1'. (Related to item 1.) Could the authors clarify the reason why θ1000 are not examined at the beginning of the analysis in the text more explicitly if the authors don't do them? Is this because "Environmental changes associated with variations in the strength of the subtropical high pressure regions correlate better with LTS (and EIS) than they do with the components of the LTS and EIS the variations are thought to influence (L240-241)"?

The reason that we didn't show the dependence of $\theta_{1000}$ on those influencing factors is that the dependence of those influencing factors on the geopotential gradient is already rejected (section 3.3). However, as both reviewers are interested in seeing that disconnection explicitly. Therefore, we added a new Fig. 8 showing the relationships between $\theta_{1000}$ and geopotential gradient, LHF, and ocean upwelling in section 3.3.

**Minor comments**

1. L1-2: "Stratocumulus (Sc) covers the eastern flanks of maritime subtropical high pressure systems and exerts an influence on the global energy budget comparable to CO2."

This sentence seems to be unclear. Do the authors mean the total CO2 radiative effect

(difference between the current CO2 concentration level and zero CO2)? Or the change of (doubling) CO2 (~4 W/m2)? The influence of Sc is (the global average of) cloud radiative effect of Sc itself (difference between the current Sc existence and clear state. not change in CRE of Sc)?

We meant: comparable to a doubling of CO2. This sentence has been corrected in L1-2.

2. L4: "gradient"
"difference" is better?
It has been replaced.

3. L39-41: "The first hypothesized mechanism would be that variations in the strength of the high influence the free-tropospheric temperature above cloud top through enhanced adiabatic warming. This mechanism follows the pioneering work of Rodwell and Hoskins (2001), who demonstrate how monsoons can influence the strength of the high to their west"

The first sentence mentions the influence of high-pressure systems on clouds (the cause is the variation in high-pressure systems and the effect is the variation in clouds). The second sentence seems to discuss the influence of the monsoons on the variation of high-pressure systems (the cause is the monsoon variations and the effect is the variations of high-pressure systems.). Are the mechanisms similar?

The connection between the first hypothesized pathway and the monsoon-desert mechanism has been clarified and this part has been rephrased in L42-46.

4. L157-164 (2nd paragraph of 3.1):
I thought the essence of these characteristics was discussed in Klein & Hartmann (1993). So it's appropriate to cite the publication here.
We've added a citation in L193-194 following the discussion of the seasonal timescale.

5. L168: "or aloft, driven by downwelling"
What is "downwelling"? Air (atmospheric) subsidence? Or ocean downwelling? Could you please clarify what downwelling or upwelling you mean when you use these terms?
It means atmospheric subsidence here. We've additionally clarified all the other uses of the terms upwelling/downwelling in the manuscript.

6. L169-171: "For the seasonal cycle, the three regions with the largest variability of the strength of the high (measured by the standard variation of ω700,H in Table 1) are also the regions where θ700 varies more than θ1000. However, this correlation does not imply causation, as the interannual series fails to exhibit the same relationship."

Especially mid-latitude (including subtropics), the seasonal cycle of θ700 is mainly controlled by the seasonal cycle of solar zenith angle (not by omega700), I thought. I guess people generally imagine that ω700 cannot determine θ700 (at least) except tropics. Am I wrong?

The Sc regions in our study are not exclusively mid-latitude: several (NP, SP, SA) extend into the tropics, while others reach higher latitudes. Therefore, we believe it worth to explicitly test the correlation between $\omega_{700}$ and $\theta_{700}$ here.

Our goal in comparing the variability of $\omega_{700}$ and $\theta_{700}$ across regions on both timescales was to test whether such a relationship might emerge statistically under our hypothesized free-tropospheric pathway. The discussion here on seasonal timescales is not the main focus; instead, it serves structurally as a lead-in to highlight the lack of correlation on interannual timescales. We have expanded the last sentence to clarify that statement in L203-206.

7. L177-179: "The somewhat weaker relationship between EIS and LHF can be expected because while surface winds cool the surface, and hence lower θ1000, increased latent heat fluxes also breakup the cloud decks"

This sentence discusses the relationship between EIS and LHF. So the fact that increased latent heat fluxes break up the cloud decks cannot be the reason for the relationship between EIS and LHF (i.e., cloud cover doesn't affect EIS directly), I guess.

Sorry for the misleading sentence. In the previous version, we implicitly treated EIS as a proxy of Sc cloudiness, which led to the inclusion of that discussion. However, we recognize that using the influence of LHF on cloudiness as support for its impact on EIS is not logically rigorous. Therefore, we have removed this sentence to avoid confusion.

8. L184: "atmospheric downwelling"

Does this mean the air (atmospheric) subsidence?

Yes, it's been changed to subsidence here.

9. Captions of Figure 3 and 6:

Plotted variables look not raw values but anomalies of the variables. In addition, I couldn't find the meaning of the apostrophe attached to the variables.

Yes, the plotted variables are anomalies there. We've added a clarification that "The primes indicate deviations from the mean of the respective regions on the corresponding timescales." in both captions.

10. L230: "cloud-top temperature"

L232: cloud-top

Generally, 700 hPa is not cloud top but above stratocumulus and in the free atmosphere, I guess.

Thanks. Yes, we automatically use the word "cloud" to represent stratocumulus in the manuscript but realized that it may be misleading. Hence, we replaced the first statement with "temperature above Sc" (L271) and the second one with "free troposphere" (L273).

11. L242: "upwelling velocity"

Can you clarify what upwelling this is? Ocean upwelling velocity?

Yes, it's ocean upwelling velocity and has been clarified. (L286)

---

## Author Comment (AC2)

Please note that the line numbers referenced in this response refer to **the version of the manuscript with tracked changes**, as they may differ from the ones in the clean version.

**Response to Referee #2**

**Minor Comments**

(1) P.3 L.65 "This paper uses the second version of the ATSR-AATSR"
Readers may be interested in the characteristics of this data set when compared with other observations such as ISCCP or MODIS. If the authors could comment on this, that will be helpful.
The ATSR-AATSR data set is popular in Europe as it's developed and operated by ESA, but we agree that adding its characteristics will be helpful. We've added a sentence in L.87-89.

(2) P.5 L.109 "... it is fixed to be 500m"
It would be helpful to readers if the authors explain why the LCL is fixed to be 500m, rather than being estimated.
LCL is estimated by a simple approximation of dew point and itself. First of all, we have to emphasise that the difference between EIS and LTS is dominated by the variability in lapse rate rather than that in LCL or the geopotential height of 700 hPa (Wood and Bretherton, 2006). Based on that we can do the simplification in calculating LCL. An additional clarification is put in L.130-134.

(3) P.5 Figure 2 caption "low cloud fraction (kappa) from 2003 to 2014"
This appears inconsistent with the statement in L.70, that low cloud fraction data for the period from Jan 2003 to Dec 2011 is analyzed.
It was a typo and we've changed it to 2011 now.

(4) P.6 L.143 "C_D=0.0015 the drag coefficient"
Readers might wonder why the drag coefficient is assumed to be a constant value of 0.0015. Additional clarification would be helpful.
The drag coefficient $C_D$ is assumed to be constant in this study, as its variability is relatively small and sensitivity to it is not our focus. Previous studies have shown that $C_D$ varies little under moderate wind conditions (<10 m/s) (Vickers, Mahrt, and Andreas, 2013; Kochanski, Koračin, and Dorman, 2006). Our analysis shows that the mean 10 m wind in the studied regions ranges from 1.7 to 7.3 m/s, with extreme values ranging from 0 to 12.2 m/s. Based on this, we initially selected a value of 0.0015 for $C_D$, referring to Fig 5 in Vickers, Mahrt, and Andreas (2013), which represents the maximum of $C_D$ within the range of extreme values.
However, following your comment, we see that this choice may lead to a debate which we wouldn't expect to. To avoid distraction, we decide to choose a classical and specific method to define $C_D$. Hence, we now changed $C_D$ to be 0.0012 referring to Large and Pond (1981). Figures 3 and 7 in the manuscript have been updated accordingly (with no significant change), and a clarification has been added to L.173–174.

(5) P.7 Table 1 caption "The dominant contribution to EIS is denoted by a bold font"
It would be helpful to readers if the authors explain in more detail what the dominant contribution means. Does it mean that magnitude of the correlation or the standard deviation at one pressure level is larger than the magnitude at the other level?
It refers to the magnitude of the correlation. We've rephrased the sentence to be "The level with the higher correlation is denoted by bold font" in the caption of Table 1.

(6) P.7 L.158 "explain most of the variability in EIS"
Readers might wonder if this statement is supported by Table 1, because magnitude of the correlation is smaller at 700hPa than at 1000hPa at NP and SI on interannual timescale. Additional clarification would be helpful.

It's a bit misleading to use the word "most" here. We've rephrased the sentence to be "In the higher latitude regions of the NP, NA and SI, variations of $\theta_{700}$ are mostly larger than variations in $\theta_{1000}$ in both the seasonal and interannual data and explain a large part of the variability in EIS in those regions, particularly on seasonal timescales".

(7) P.7 L.171 "this correlation does not imply causation"
Here the authors argue that the correlation does not imply causation. Does this argument apply to both seasonal cycle and interannual variability, or does it apply to interannual variability only? Additional clarification would be helpful.

I've phrased it in L.203-206.

(8) P.8 Figure 3
I suggest that the authors give definition of the primes for EIS, Q_c, LHF_c, and W_E.

I've added an explanation that "The primes indicate deviations from the mean of the respective regions on the corresponding timescales." in the caption.

(9) P.8 L.187 "a clear relationship between Q_c and omega_700,c"
It would be helpful if the authors write the pressure level at which the Q_c is evaluated. Is it 700hPa?

Yes, it's 700 hPa.  It's been added here (L.222).

(10) P.10 L.204 "Upwelling areas are restricted to the coastal regions where the wind-stress curl is large"
In Figure 7(b), there appears to be no upwelling along the equator in the Pacific and the Atlantic. Readers might wonder how the SST cold tongue is maintained. Additional clarification would be helpful.

We cannot show the upwelling along the equator due to the change of sign in the Coriolis parameter as the Ekman pumping velocity in this paper is calculated from wind stress. Referring to equation 7, $w_E$ cannot be properly calculated near the equator where $f$ is approaching zero. Hence, we masked the values in the current Figure 7(b) near the equator but it's not explicit as it is covered by the gridline. Hence, we remove gridlines, extend the masked region to avoid misleading, and state it in the caption of the current Figure 7.

(11) P.10 L.207 "variations in near-surface geopotential gradients are not the primary driver of changes in theta_1000"
It may be a good idea to show scatter plots for variations in geopotential gradients and variations in theta_1000, for both seasonal and interannual timescales, so that readers can better follow the argument.

I've added a new Figure 8 showing the variation relationships between geopotential gradients and $\theta_{1000}$ (the left column).

**Typos**
Thanks for pointing them out. All the typos mentioned here have been corrected.

---

## Referee Report (RR1)

I appreciate the author's efforts in revising the manuscript.

Now, many parts have been clarified. And thank you for adding several data and plots. They are very useful.

Now I recommend acceptance of the manuscript.

I have one suggestion.

If the authors add a schematic figure of the variables whose correlations are discussed in the paper, it would be very helpful and useful for readers to understand and refer to the results, because many relationships are shown and discussed. But this is just a suggestion, and it's completely up to the author's preference.

Example of a schematic figure:

LCF $\longleftrightarrow$ EIS $\longleftrightarrow$ $\theta$ 700 $\longleftrightarrow$ Q $\longleftrightarrow$ $\omega$ 700 $\longleftrightarrow$ SLP

EIS $\longleftrightarrow$ $\theta$ 1000 $\longleftrightarrow$ ocean upwelling $\longleftrightarrow$ SLP gradient $\longleftrightarrow$ SLP $\longleftrightarrow$ latent heat flux $\longleftrightarrow$

Over $\longleftrightarrow$, corresponding figure numbers and/or table numbers are added.

In addition, if possible, the results of correlation (high or low or something) are added.

This schematic figure can be at the beginning of the manuscript or in the middle of or after the results section, and either in the main text or as a figure in supplemental information.

This is just an example (the variables could be different).

---

## Author Response (AR2)

I greatly appreciate the suggestion.

An additional schematic plot has been added to the conclusion section, labeled as Figure 11. Please refer to the updated manuscript.